# IL-8 Secreted by Gastric Epithelial Cells Infected with *Helicobacter pylori* CagA Positive Strains Is a Chemoattractant for Epstein–Barr Virus Infected B Lymphocytes

**DOI:** 10.3390/v15030651

**Published:** 2023-02-28

**Authors:** Diana A. Domínguez-Martínez, José I. Fontes-Lemus, Alejandro García-Regalado, Ángel Juárez-Flores, Ezequiel M. Fuentes-Pananá

**Affiliations:** 1Research Unit on Virology and Cancer, Children’s Hospital of Mexico Federico Gómez, Mexico City 06720, Mexico; 2Unidad de Investigación Médica en Inmunoquímica, Hospital de Especialidades, Centro Médico Nacional Siglo XXI, Instituto Mexicano del Seguro Social, Ciudad de México 06720, Mexico

**Keywords:** Epstein–Barr virus (EBV), *Helicobacter pylori* (*H. pylori*), CagA, chemoattraction, IL-8, CXCR1, CXCR2, B lymphocytes

## Abstract

*Helicobacter pylori* and EBV are considered the main risk factors in developing gastric cancer. Both pathogens establish life-lasting infections and both are considered carcinogenic in humans. Different lines of evidence support that both pathogens cooperate to damage the gastric mucosa. *Helicobacter pylori* CagA positive virulent strains induce the gastric epithelial cells to secrete IL-8, which is a potent chemoattractant for neutrophils and one of the most important chemokines for the bacterium-induced chronic gastric inflammation. EBV is a lymphotropic virus that persists in memory B cells. The mechanism by which EBV reaches, infects and persists in the gastric epithelium is not presently understood. In this study, we assessed whether *Helicobacter pylori* infection would facilitate the chemoattraction of EBV-infected B lymphocytes. We identified IL-8 as a powerful chemoattractant for EBV-infected B lymphocytes, and CXCR2 as the main IL-8 receptor whose expression is induced by the EBV in infected B lymphocytes. The inhibition of expression and/or function of IL-8 and CXCR2 reduced the ERK1/2 and p38 MAPK signaling and the chemoattraction of EBV-infected B lymphocytes. We propose that IL-8 at least partially explains the arrival of EBV-infected B lymphocytes to the gastric mucosa, and that this illustrates a mechanism of interaction between *Helicobacter pylori* and EBV.

## 1. Introduction

With about 1.1 million new cases and more than 750,000 deaths per year worldwide, gastric cancer (GC) is a major health problem, particularly in areas of high incidence, such as Asian and Latin American countries [1]. GC is well known for its infectious etiology in which *Helicobacter pylori* (*H. pylori*) and the Epstein–Barr virus (EBV) are considered the main risk factors [2,3]. *H. pylori* colonizes the gastric epithelium of at least half of the world’s population. The persistence of *H. pylori* results in chronic inflammation associated with the development of gastric disease, including chronic gastritis, ulcer diseases and GC [4]. EBV also causes persistent infections in approximately 90% of the adult population worldwide, and it is estimated that nearly 10% of GCs are linked to a direct EBV infection [5,6]. Both pathogens are considered human oncogenic agents by the International Agency for Research on Cancer (IARC) [2,3].

*H. pylori* is a Gram-negative bacterium that expresses a battery of virulence genes that facilitates its persistence in the low pH of the gastric environment. The *cag* (cytotoxin-associated gene) pathogenicity island (*cag*PAI) is a major virulence factor that encodes the components of a type IV secretion system and the CagA effector protein (cytotoxin-associated gene A). The cagPAI propels the CagA injection into the host’s gastric epithelial cells [7,8]. In the cell’s cytoplasm, CagA activates signaling pathways that lead in vitro to the remodeling of the cell cytoskeleton, and acquisition of an elongated morphology known as the hummingbird phenotype [9,10]. CagA also induces a loss of the cell´s apico-basal polarity through altering the function of the PAR1/MARK kinase [11,12]. CagA and the cagPAI also induce the gastric epithelial cells to secrete IL-8 upon the activation of nuclear factor kappa B (NF-κB) complexes. IL-8 is a potent chemoattractant for neutrophils and one of the most important chemokines for the *H. pylori*-induced chronic gastric inflammation [13]. High densities of neutrophil infiltration mark active gastric lesions according to the Sydney classification [14,15,16]. IL-8 stimulation activates CXCR1 and CXCR2 receptors and signaling pathways related to the mitogen-activated protein kinase pathway (p38MAPK/Erk) [8,11,17,18].

EBV is transmitted by saliva and infects naïve B lymphocytes present in the lymph nodes of the oro-naso-pharynx. It is proposed that EBV persists in memory B lymphocytes in immunocompetent individuals, and more likely in secondary lymphoid organs, in a latent state in which there is not a production of viral infectious particles [19]. However, an infection of the epithelial cells of the oro-naso-pharynx has also been demonstrated [20,21,22]. In vitro and in vivo models have documented a preferred lytic replication in epithelial cells [23]. An infection of the oro-naso-pharynx epithelial cells is proposed to facilitate the release of EBV particles into saliva and its transmission to new hosts [24]. The infection of these epithelial cells probably occurs through the virus released by reactivated B lymphocytes residing in the lymph nodes of the Waldeyer´s ring [19,25,26].

We have previously documented a correlation between *H. pylori* and EBV co-infection and severe gastric inflammation [27,28,29,30,31]. In those studies, higher levels of antibodies against the EBV lytic protein VCA (viral capsid protein) marked more advanced gastric lesions and a more abundant infiltrate of mononuclear and polymorphonuclear immune cells. In our studies, EBV seemed to be contributing to gastric inflammation from very early lesions, including pediatric gastritis [30]. However, to date, the mechanism by which EBV reaches, infects and persists in the gastric epithelium is not understood. Since *H. pylori* is the obligated resident of the stomach, in this study we assessed whether the *H. pylori* infection would facilitate the chemoattraction of EBV-infected B lymphocytes. We identified IL-8 as a powerful chemoattractant produced in gastric epithelial cells in response to the bacterial infection, particularly in response to an infection with the most virulent CagA positive *H. pylori* strains. We also identified CXCR2 as the main IL-8 receptor, whose expression is induced by EBV on infected B lymphocytes.

## 2. Materials and Methods

### 2.1. Helicobacter pylori and Cell Cultures

We worked with two *H. pylori* CagA(+) strains in this study: strain NCTC11637 (ATCC No. 435049TM, Manassas, VA, USA) with a Western-type CagA (EPIYA ABCCC), referred to throughout the text as CagA(+) ABCCC; and strain NY02-149 with an East-Asian-type CagA (EPIYA ABD), referred to as CagA(+) ABD, which was kindly donated by Dr. Guillermo Perez-Perez from New York University. We additionally worked with two *H. pylori* CagA(−) strains: strain 365A3, which has a *cag*PAI lacking the whole effector protein CagA (CagA(−)), and strain 254A3 with a nonfunctional *cag*PAI (CagPAI(−)). These were kindly donated by Dr. Javier Torres and Dra. Margarita Camorlinga from Instituto Mexicano del Seguro Social (IMSS) [32]. All *H. pylori* strains were grown on blood agar (Sigma-Aldrich, St Louis, MO, USA) for 22–24  h at 9% CO_2_ and 37 °C.

The following cell lines were used in this study: AGS (No. CRL-1739), NCI-N87 (No. CRL-5822) and Kato-III (No. HTB-103) human GC cell lines; Hs445 (No. HTB-146), B95-8 (No. CRL-1612) and Ramos (No. CRL-1596) human B lymphocyte cell lines. All cell lines were obtained from the ATCC (Manassas, VA, USA). The Akata cell line (GFP-EBV positive, latency I) was kindly donated by Kenzo Takada from Hokkaido University in Japan [33]. NCI-N87, B95-8, Akata, Hs445 and Ramos cell lines were cultured in RPMI-1640 medium, AGS cells were cultured in F12 medium and Kato-III cells were cultured in IMDM medium. RPMI-1640 and F12 cell culture media were purchased from Life Technologies (Grand Island, NY, USA), and IMDM cell culture media was obtained from ATCC (Manassas, VA, USA). RPMI-1640, F12 and IMDM cell cultures were supplemented with 10% heat-inactivated fetal bovine serum (FBS) and antibiotic/antimycotics (100 U/mL penicillin, 100 μg/mL streptomycin and 0.25 μg/mL amphotericin B (Gibco, Carlsbad, CA, USA)). The Akata cell line was cultured in RPMI-1640 containing G418 (700 µg/mL, Sigma-Aldrich, St. Louis, MO, USA) to maintain the EBV bacmid. All cell cultures were maintained at 37 °C in humidified air and 5% CO_2_.

### 2.2. Purification of Primary Blood B Lymphocytes (PBLs)

B lymphocytes were obtained from peripheral blood mononuclear cells (PBMCs) of three different healthy donors. PBMCs were isolated by differential separation using Histopaque 1077 (Sigma-Aldrich, St. Louis, MO, USA), and the B lymphocyte-enriched fraction was obtained by a negative selection using the B cell Isolation Kit-II (Miltenyi Biotec, Bergisch Gladbach, Germany) according to the manufacturer’s instructions. The B lymphocyte purity was assessed by the flow cytometry. PBLs were activated with 100 ng/mL of Phorbol 12-Myristate 13-Acetate (PMA, Sigma-Aldrich, St. Louis, MO, USA) for 4 h, then cells were washed twice with Phosphate Buffered Solution 1x (PBS, Life Technologies, Grand Island, NY, USA); finally, activated PBLs were used as positive controls for migration assays using FBS as a chemoattractant. The study was approved by the Ethical, Biosafety, and Scientific Institutional Review Boards of the Children’s Hospital of Mexico “Federico Gómez” (HIM-2017-074), and the individuals willing to participate signed a consent form.

### 2.3. Infection of Gastric Epithelial Cells and Generation of Conditioned Media

To obtain conditioned media from gastric epithelial cells infected with *H. pylori* strains, GC cell lines were cultured in 12-well plates (Corning, Corning, NY, USA) in their respective medium at 1.5 × 10^5^ cells per mL. *H. pylori* colonies were harvested and resuspended in PBS 1x (Life Technologies, Grand Island, NY, USA) to quantify the bacterial density at 550 nm. Bacterial suspensions were adjusted to infect the GC cell lines at a multiplicity of infection of 100 in their respective serum-free medium for 6 h at 5% of CO_2_ and 37 °C. *H. pylori* was then eliminated by treating with gentamicin (300 ng/mL; Sigma-Aldrich, St. Louis, MO, USA) for 1 h; the GC cell lines were washed twice with PBS 1x and maintained for 18 h in their respective serum-free medium to complete 24 h at 5% of CO_2_ and 37 °C. Then, conditioned media were collected, centrifuged at 1500 rpm/5 min to eliminate floating cells, and used immediately to evaluate the chemoattraction of the different human B lymphocyte cell lines and for the IL-8 quantification.

#### 2.3.1. IL-8 Quantification

The concentration of IL-8 in all the conditioned media obtained from the *H. pylori*-infected GC cell lines was measured by an enzyme-linked immunosorbent assay (ELISA), using an OptEIA human IL-8 (BD Biosciences, Piscataway, NJ, USA) and following the manufacturer’s instructions.

#### 2.3.2. Chemoattraction Assays: Migration and Invasion

One hundred fifty thousand B lymphocytes were used per experimental condition. Cells were resuspended in 150 μL of F12 serum-free medium and placed in the upper chamber of a transwell insert of 6.5 mm diameter and 5 μm of pore size (Corning, Kennebunk, ME, USA). For invasion assays, Matrigel (Corning) was added to the transwell insert diluted in a F12 serum-free medium (1:3); for migration assays, transwell inserts were untreated. Transwells were placed in 24-well culture dishes containing 800 μL of conditioned media obtained as in Section 2.3, or with medium supplemented with IL-8 recombinant protein (rIL-8, Miltenyi Biotec, Auburn, CA, USA) as the chemoattractant. The concentration of the rIL-8 employed was 2200 pg/mL, which is the highest level of IL-8 found in the conditioned media of GC cell lines infected with *H. pylori* CagA(+) strains. In addition, we performed an invasion kinetic assay of B lymphocytes (Akata cell line) using increased concentrations of rIL-8: 225 pg/mL, 550 pg/mL, 1100 pg/mL, 2200 pg/mL and 4400 pg/mL. For each experiment, the F12 basal medium was used as a negative control, whereas the culture medium supplemented with 6% of FBS was used as a positive control. B lymphocytes were then left for 24 h for migration assays or 48 h for invasion assays at 37 °C in a humidified 5% CO_2_ environment. Migrating or invading cells were observed in the bottom wells using a digital camera, Motic 5.0 MP, and the Motic image plus 3.0 software (Motic, Kowloon, Hong Kong). A total of 5 fields/well in 10× magnification were captured and counted. Three independent assays were performed for each assay.

### 2.4. CagA Overexpression in AGS Cells

Plasmids encoding full-length GFP-CagA (referred in the text as GFP-CagA1-1216), and full-length CagA (referred in the text as CagA1-1216) were kindly provided by Dr. Manuel R. Amieva (Stanford University School of Medicine) [34]. The empty vector (pGFP-C1; Clontech) was used as a control. AGS cells were grown to 70% confluence in coverslips in 6-well plates and transfected with Lipofectamine 3000 (Invitrogen Life Technologies, MA, USA) according to the manufacturer’s indications. The total amount of plasmid DNA in transfections was 2.5 µg/per well. The transfected cells were maintained in F12 serum-free medium (Gibco, Carlsbad, CA, USA). Twenty-eight hours post-transfection, the conditioned media were collected and centrifuged; IL-8 levels were detected (see Section 2.3.1.) and functional assays (chemoattraction assays with the Akata cell line) were conducted, as previously explained in Section 2.3.2.

#### Immunofluorescence

Transfected AGS cells were fixed with 4% paraformaldehyde in PBS 1x (Gibco, Carlsbad, CA, USA) for 20 min at room temperature and washed three times with PBS 1x. The cells on coverslips were mounted on glass slides using Vectashield (Vector Laboratories, Newark, CA, USA). AGS cells transfected with GFP-plasmids were directly observed with the microscope, while AGS cells transfected with the CagA1-1216 construct, after fixation, were blocked with 1% BSA in PBS 1x for 30 min, then incubated with mouse anti-CagA (Sc-28368; Santa Cruz, CA, USA) for 1 h at 37°C, followed by incubation with anti-mouse Alexa Fluor 647 (#715-605-150; Jackson ImmunoResearch Laboratories, West Grove, PA, USA) for 40 min at room temperature. Cells on coverslips were mounted on glass slides using Vectashield.. Images were acquired with a Nikon Eclipse Ti confocal laser scanning microscope (Nikon Corp) with a 20x objective.

### 2.5. Western Blot Assays

The following antibodies were used for the Western blot assays: mouse anti-phospho-ERK1/2 (p-ERK1/2, Thr202/tyr204) (GTX01452), rabbit anti-ERK1/2 (ERK1/2, GTX59618), rabbit anti-phospho-p38 MAPK (p-p38, Thr180/Tyr183) (GTX33599) and rabbit anti-p38 MAPK (p-38, GTX33621); all antibodies are from GeneTex Inc. (Irvine, CA, USA). β-Actin-HRP (Abcam, ab49900; Cambridge, UK) and HRP-linked goat anti-mouse (626520) or HRP-linked goat anti-rabbit (31460) (ThermoFisher Scientific, Rockford, IL, USA) were used as secondary antibodies.

Akata or Ramos cells were seeded in a 25 cm^2^ culture flask at a density of 3 × 10^6^ per flask; the cells were serum-starved for 18 h and then treated for 5 min with 2200 pg/mL of rIL-8. In some experiments, cells were treated with 400 nM of SB225002 (Tocris Bioscience, Bristol, UK) or reparixin (Sigma-Aldrich, St. Louis, MO, USA) during 4 h to block CXCR1 or CXCR2 signaling before the rIL-8 treatment. The total proteins were extracted from cells using a RIPA lysis buffer (20-188; EMD Millipore, Darmstadt, Germany) supplemented with 1 mM phenylmethylsulfonylfluoride, 1x protease inhibitor cocktail I and phosphatase inhibitors (1 mM sodium vanadate, 1 mM sodium fluoride and 10 mM β-glycerolphosphate) from Sigma-Aldrich, St. Louis, MO, USA.

The protein extracts were forced through a 22-gauge needle 10 times and centrifuged for 10 min at 14,000 rpm at 4 °C. The protein concentration was determined by the 2-D Quant kit (Cytiva, 80-6483-56 Sigma-Aldrich, St. Louis, MO, USA). The protein samples were mixed with 4x Laemmli buffer (3:1) and heated at 90 °C for 5 min. Equal amounts of protein (30 μg/sample) were resolved in the 10% SDS-PAGE under reducing conditions, then transferred to a PVDF membrane (Millipore, Billerica, MA, USA). Non-specific binding to the membrane was blocked with Tris-buffered saline (10 mM Tris-HCl, pH 8.0, 150 mM NaCl, 5% (*w*/*v*) low-fat dry milk and 0.05% (*v*/*v*) Tween-20) for 1 h at room temperature. Then, the membrane was incubated with the corresponding specific primary antibody for 2 h at room temperature, and followed by the appropriate HRP-conjugated secondary antibody. The HRP antibody binding was detected using Immobilon Forte (Millipore, Billerica, MA, USA). Finally, the membranes were visualized with a ChemiDoc MP imaging system (Bio-Rad, Hercules, CA, USA).

### 2.6. Quantitative Real-Time PCR Assays

*CXCR1* and *CXCR2* were quantified in Akata and Ramos cell lines in basal conditions. In other experiments, the *IL-8* temporal gene expression was analyzed on AGS cells infected with the same strains of *H. pylori* at 4, 6, 8 and 12 h. Total RNA from Akata, Ramos and AGS cells was extracted using the RNeasy mini kit (Qiagen, Valencia, CA, USA) according to the manufacturer’s instructions. The cDNA was synthesized from 1 µg of the total RNA using the qMax First Strand cDNA Synthesis Flex kit (Accuris Instruments, Edison, NJ, USA). Quantitative PCR (qPCR) was performed using the Maxima SYBR Green/Rox qPCR Master Mix (x2) (Thermo Fisher, Dreieich, Germany). The primers’ sequences were as follows: CXCR2-F: 5′-GCTCTGACTACCACCCAACCTTGA-3′, CXCR2-R: 5′-AGAAGAGCAGCTGTGACCTGCTGT-3′; CXCR1-F: 5′-CCTGGCCGGTGCTTCAGTTA-3′, CXCR1-R: 5′-ATCAAAATCCCACATCTGTGGATCT-3′; IL-8-F: 5′-CACCGGAAGGAACCATCTCACTGT-3′, IL-8-R: 5′-TCCTTGGCAAAACTGCACCTTCA-3′; and GAPDH-F: -5′-CTTCACCACCATGGAGAAGGC-3′, GAPDH-R: 5′-GGCATGGACTGTGGTCATGAG-3´. The thermal cycling conditions started with an initial denaturation step at 95 °C for 10 min, followed by 50 cycles at 95 °C for 15 s and 65 °C for 1 min. The relative expression of mRNA was calculated by the ^2−ΔΔCt^ method and normalized to GAPDH.

### 2.7. IL-8 Knockdown

The siRNAs against human IL-8 (IL-8-siRNA) and control scramble siRNA-A were purchased commercially from Santa Cruz Biotechnology (Santa Cruz, CA, USA). AGS cells were transfected with siRNAs (at a final concentration of 6 nM) using Lipofectamine 3000 (Invitrogen Life Technologies). For this, AGS cells were grown to 70% confluence in 12-well plates and transfected the following day according to the manufacturer’s instructions. First, Lipofectamine 3000 and Opti-MEM medium (Gibco, Carlsbad, CA, USA) were mixed, and in another mixture, either siRNA, Opti-MEM, and Lipofectamine 3000 Reagent were mixed for 5 min. The two mixtures were then combined and incubated for 15 min at room temperature, and finally, 100 µL of the solution was added to each well. The transfection was complete after 24 h; then, the AGS cells were infected with *H. pylori* strains as described in Section 2.3. After 24 h of additional incubation, the resulting conditioned media was collected and centrifuged at 1500 rpm for 5 min at 4 °C and tested in B lymphocytes invasion assays. The *IL-8* knockdown was confirmed by ELISA as explained in Section 2.3.1.

### 2.8. Bioinformatics Analysis

RNA-seq Genomic Data Commons (GDC) TCGA (The Cancer Genome Atlas) gene expression data was retrieved from the gastric cancer XENA platform of the University of California Santa Cruz (UCSC) data, with a total of 240 samples divided into: 218 EBV positive samples (EBVaGC) and 22 EBV negative samples (EBVnGC). The R language was used to transform the UCSC data to obtain the raw counts. At the same time, a list of the original GDC TCGA GC study was created. The list was used to filter the data of samples obtained from UCSC. Cloud computing with useagalaxy.org was used for a differential gene expression (DGE) analysis between EBVaGC and EBVnGC samples using an edgeR tool with default parameters previously reported by the TCGA Research Network [35,36]. The resulting genes from DGE with a fold-change of at least 2.5 and a false discovery rate (FDR) with a *p* value **≤** 0.05 were uploaded into the WebGestalt bioinformatics web tool [37] to perform an Over-Representation Analysis for Gene Ontology (GO). A normalized gene enrichment analysis (NES) was performed using normalized gene expression data along with a list of genes of interest (see Appendix A, [16,38,39,40,41,42,43,44,45,46,47,48,49,50,51,52,53,54]) and a Phenotest package [55]. A Kaplan–Meier survival curve analysis was made from 220 samples of GC with available clinical information. The survival analysis was performed comparing between groups with high and low expressions of *IL-8*, *CXCR1* and *CXCR2*. For this analysis, a cutoff of 40 months and a z-score of >1 or <−1 were used. Lubridate, survival and survminer libraries were used for the Kaplan–Meier analysis [56,57].

### 2.9. Statistical Analysis

Statistical analysis was performed using the GraphPad Prism v8.0.1 software (GraphPad, San Diego, CA, USA). We first analyzed the data distribution using the D’Agostino and Pearson test. For data in which more than two groups had a normal distribution, the one-way analysis of variance (ANOVA) or two-ANOVA was performed. Data lacking a normality and/or homogeneity of variance were analyzed with the non-parametric Kruskal–Wallis and Dunn’s test for multiple comparisons as a post hoc assessment. The specific statistical analysis performed is indicated in the description of each figure legend. Descriptions of experimental replicates are also described in figure legends. The results are shown as the mean ± standard error of the mean (SEM). *p* values ≤ 0.05 were considered statistically significant.

## 3. Results

### 3.1. EBV-Infected B Lymphocytes Are Chemoattracted by H. pylori CagA(+) Strains

In order to assess whether *H. pylori* infection could facilitate the chemoattraction of EBV-infected B lymphocytes, we explored whether components present in conditioned media from gastric epithelial cells infected with *H. pylori* promote the chemoattraction of EBV-infected B lymphocytes. We infected AGS cells with *H. pylori* strains and tested the supernatants (conditioned media) as chemoattractants of Akata cells (EBV-GFP positive, latency I), and performed migration and invasion assays. We found that Akata cells were chemoattracted when we used the conditioned media from AGS cells infected with *H. pylori* CagA(+) strains. Figure 1A shows representative fluorescence images of chemoattracted EBV-GFP positive B cells. Figure 1B,C show the plots of the mean numbers of three independent replicates of migration and invasion assays, respectively, in which it can be observed that *H. pylori* CagA(+) ABD and ABCCC strains were significantly superior (*p* < 0.0001) than CagA(−) and CagPAI(−) strains. We confirmed the capacity of the conditioned media from AGS cells infected with *H. pylori* CagA(+) strains to chemoattract EBV-infected B lymphocytes, and also tested the EBV positive Hs445 (latency II) (Figure 1D) and B95-8 (latency III) B cell lines (Appendix A). Conditioned media derived from the following gastric cell lines infected with the *H. pylori* CagA(+) ABD strain: Kato-III (Figure 1E) and NCI-N87 (Appendix A) also favored the chemoattraction of EBV-infected B lymphocytes (Akata cell line). We verified the viability of the Akata cells maintained in the conditioned media from AGS cells infected with *H. pylori* strains CagA(−) and CagA(+) ABD. We observed that less than 9% of Akata cells died during the time that the chemoattraction assays lasted (Appendix A).

### 3.2. CagA Overexpression in AGS Cell Promotes the Invasion of EBV-Infected B Lymphocytes

Since our data suggest that the CagA protein may be important to mediate the chemoattraction of EBV-infected B lymphocytes, we assessed whether the CagA protein overexpressed in AGS cells promoted the release of biomolecules into the supernatant that favors the chemoattraction of EBV-infected B lymphocytes. We transfected plasmids encoding full-length GFP-CagA1-1216 and CagA1-1216 in AGS cells (Figure 2A and Appendix A). Twenty-eight hours post-transfection, the supernatants were harvested and tested for invasion assays with the Akata cell line. We observed that only the components of the conditioned media of AGS cells overexpressing the full-length CagAs were able of promote the invasion of Akata cells compared with the AGS cell untransfected or transfected with the empty vector, pGFP-C1 (Figure 2A,B). We noticed that the number of invading Akata cells in response to the CagA overexpression was lower than the number observed with the *H. pylori* infection (Figure 1C), suggesting that other bacterial or cellular components released during the gastric epithelial cell infection may also be contributing to the chemoattraction of EBV-infected B lymphocytes. Appendix A shows the viability of AGS cells. The percentages of dying AGS cells transfected with GFP-CagA and CagA plasmids were less than 3% by the time the conditioned media were harvested. The viability of Akata cells maintained in the conditioned media obtained from transfected AGS cells was above 96% (Appendix A).

### 3.3. Active EBV and H. pylori Infection Are Necessary for B Lymphocyte Chemoattraction

We assessed whether the chemoattraction towards gastric epithelial cells infected with *H. pylori* CagA(+) strains was favored by the EBV infection. We tested Ramos cells, an EBV negative B cell line derived from a patient with Burkitt’s lymphoma. We were surprised to find that Ramos cells were not chemoattracted (Figure 3A and Appendix A). We confirmed this observation using purified PBLs derived from healthy donors (Figure 3B and Appendix A). It is well accepted that the EBV infection mimics the antigenic stimulation of B lymphocytes [19]. We assessed whether the activation of PBLs with PMA could promote a chemoattraction in a similar manner as the EBV infection. Treatment with PMA activates protein kinase C, a central kinase downstream of the antigenic stimulation. We observed that the PMA-stimulated PBLs was migratory to FBS (Figure 3C).

We also addressed whether an active *H. pylori* infection of gastric epithelial cells was necessary. For this analysis, the *H. pylori* strains were killed with the antibiotic gentamicin, and we carried out a co-culture of AGS cells with the dead bacteria. We thought that this was important for recognizing the chemoattraction mediated by pathogen-associated molecular pattern (PAMP) molecules that would also be present in the dead bacteria. We did not observe the chemoattraction of Akata cells in this condition, confirming that an active infection with *H. pylori* CagA(+) strains was necessary for AGS cells to actively secrete the biomolecules responsible for chemoattracting EBV-infected B lymphocytes. Appendix A shows that the *H. pylori* infection of AGS cells did not significantly kill the gastric cells during the time of the assay, as we observed less than 3% of 7-AAD positive cells. This result also reduces the possibility that the chemoattraction could be explained by the non-specific release of the damage-associated molecular pattern (DAMP) molecules. Altogether, these data support that an active infection of gastric epithelial cells by virulent *H. pylori* CagA(+) strains would attract EBV-infected B lymphocytes to the site of infection.

### 3.4. IL-8 Is an Important Chemoattractant for EBV-Infected B Lymphocytes

TCGA harbors a database of GC patients that are clearly identified as EBVaGC and EBVnGC [5]. In order to identify the candidate molecules responsible for the chemoattraction of EBV-infected B lymphocytes, we performed a differential gene expression analysis of the transcriptomic data comparing the EBVaGC versus the EBVnGC samples (Figure 4A). We found 178 genes up-regulated and 364 genes down-regulated in the EBVaGC, considering genes as up-regulated or down-regulated as those with a false discovery rate (FDR) of *p*-value ≤ 0.05. We performed a Gene Ontology (GO) enrichment analysis with the set of up- and down-regulated genes (Figure 4B,C). Interestingly, this analysis revealed that patients with EBVaGC had an enrichment of genes associated with a response to chemokines as the top process (Figure 4B). Thus, we coined a gene signature of chemokines and chemokine receptors (see Appendix A) that are increased during a *H. pylori* infection across different gastric diseases, and estimated the Normalized Enrichment Score (NES) of the signature in the EBVaGC (Figure 4D). This analysis unveiled those members of the signature with the strongest association.

We observed IL-8 among the chemokines with the highest score. IL-8 is a chemokine that triggers the migration and activation of neutrophils during the *H. pylori* chronic infection of the gastric mucosa [13,14,15], and is a chemokine important in the development of the *H. pylori*-induced gastric lesions, such as peptic ulcers, and early chronic gastritis to late cancerous lesions [13,38,39,40,58,59]. However, its role as a chemoattractant of B lymphocytes is not well-documented. We performed a kinetic analysis of the *IL-8* expression on AGS cells infected with *H. pylori* strains, confirming *IL-8* upregulation and a peak of expression at 6 h post-infection with the CagA(+) ABD strain (Appendix A). We also confirmed the presence of IL-8 in the conditioned media of the *H. pylori*-infected AGS, Kato-III and NCI-N87 gastric epithelial cell lines (Appendix A). As previously documented, IL-8 levels detected in the infection with CagA(+) strains were higher than those of CagA(−) strains, and correlated with the highest chemoattraction of EBV-positive B cell lines (Figure 1B–E; Appendix A). We also confirmed the secretion of IL-8 by AGS cells overexpressing full-length CagAs (Appendix A). It has already been reported that the IL-8 production in gastric epithelial cell lines (AGS and GES) is induced by CagA [60,61].

To determine whether IL-8 was participating in the chemoattraction of EBV-infected B lymphocytes, we performed invasion assays using different concentrations of a recombinant IL-8 (rIL-8) protein. This analysis showed that the number of invading Akata cells was proportional to the concentration of rIL-8 (Figure 4E). On the contrary, the EBV negative Ramos cell line was not chemoattracted by rIL-8 even at high concentrations (Appendix A).

The role of IL-8 as an important chemoattractant of EBV-infected B lymphocytes was confirmed through silencing assays. We transfected AGS cells with a specific siRNA targeting human *IL-8* (IL-8-siRNA) and a scramble siRNA-A control. After silencing, AGS cells were infected with *H. pylori* strains and the conditioned media were used as a chemoattractant of Akata cells. Appendix A shows the decreased concentration of IL-8 obtained in the conditioned media of the AGS cells transfected with the specific IL-8-siRNA. The reduction was of approximately 70% in cells infected with the CagA(+) ABD strain. We observed a reduction of approximately 65% in the capacity of the *IL-8*-silenced AGS conditioned media to chemoattract Akata cells (Figure 4F). We further explored whether *IL-8* expression levels in the TCGA transcriptomic database would influence the survival of patients with GC, without finding an association (Appendix A).

### 3.5. CXCR2 Mediates Chemoattraction of EBV-Infected B Lymphocytes in Response to IL-8

Since IL-8 is a potent chemoattractant for EBV-infected B lymphocytes, and the biological effects of IL-8 are mediated through the binding with two cell-surface G protein–coupled receptors, termed CXCR1 [62] and CXCR2 [63], we analyzed the CXCR1 and CXCR2 expression in Akata and Ramos cell lines. The *CXCR1* expression in Akata cells was low, and undetectable in Ramos cells, while Akata cells expressed more *CXCR2* than Ramos cells (Figure 5A), and predominantly expressed a *CXCR2* receptor (Figure 5B). We validated these data using the online transcriptomic database of Mrozek-Gorska, et al. [64], which contains transcriptomic data of naïve B lymphocytes infected with EBV during the first two weeks of infection. In this online database, the *CXCR1* expression was also scarce and did not change during the time course of the infection (Appendix A), while the *CXCR2* expression was expressed basally and further induced by the EBV infection (Appendix A). We also explored whether in the TCGA transcriptomic database the levels of the *CXCR1* or *CXCR2* expression influenced the survival of patients with GC, and found no association (Appendix A).

We evaluated the effect of inhibiting a CXCR2 activation on Akata cells chemoattraction, employing two selective inhibitors SB225002 and reparixin, both of which antagonize the binding of IL-8 to CXCR2, inhibiting its downstream signaling [65,66,67]. We observed that upon treatment with the inhibitors, Akata cells showed significantly reduced invasion into the conditioned media obtained from AGS cells infected with the *H. pylori* CagA(+) ABD strain (Figure 5C, Appendix A). The inhibitors also significantly reduced the chemoattraction of Akata cells induced by rIL-8. Figure 5C shows a heatmap that summarizes all the results obtained with the inhibitors, and Appendix A shows the reduced Akata cell chemoattraction as an example of the data obtained with the selective inhibitors.

Since CXCR2 signals through mitogen-activated protein kinases (MAPK), we also analyzed the activation of ERK1/2 and p38 MAPKs, monitoring its phosphorylation state by the Western blot in Akata and Ramos cells after treatment with rIL-8 (Figure 5D,E, Appendix A). We observed a phosphorylation of ERK1/2 and p38MAPK on Akata cells treated with rIL-8, whereas the pretreatment with the CXCR2 inhibitors demonstrated its phosphorylation (Figure 5D). In contrast, Ramos cells treated with rIL-8 demonstrated the absence of ERK1/2 phosphorylation and a modest phosphorylation of p38MAPK (Figure 5E). The treatment of the Ramos cell line with FBS did not induce ERK1/2 or p38MAPK phosphorylation. Altogether, these data support that EBV induces the expression of CXCR2 on the infected B lymphocytes. EBV-infected B lymphocytes then acquire the capacity to migrate to the gastric epithelium infected with *H. pylori* CagA(+) strains in response to the *H. pylori*-induced secretion of IL-8.

## 4. Discussion

In the present study, we demonstrated for the first time a mechanism of the chemoattraction of EBV-infected B lymphocytes mediated by IL-8, a cytokine generated during the infection of gastric epithelial cells with pathogenic *H. pylori* CagA(+) strains [68]. In addition, we confirmed the direct participation of CagA in the secretion of IL-8 by the *H. pylori* infected AGS cells, as it has been previously reported [60,61], and in the chemoattraction of the EBV-infected B lymphocytes. Furthermore, we identified CXCR2 as an important receptor mediating the IL-8 response in the EBV-infected B lymphocytes. Interestingly, CXCR2 seems to be up-regulated in response to the EBV infection itself, since EBV-negative and naïve B lymphocytes do not migrate in response to rIL-8 or *H. pylori* infection. Our results suggest a plausible scenario in which the migration of EBV-infected B lymphocytes to the stomach is facilitated by the *H. pylori*-induced inflammatory response, which is characterized by high levels of IL-8, particularly upon infection with the most virulent *H. pylori* CagA(+) strains.

Importantly, the chemoattraction of EBV-infected B lymphocytes is not fully explained by IL-8. The supernatants of *H. pylori*-infected gastric cells have a superior chemotactic activity than rIL-8, and the IL-8 inhibitors did not completely abolish the migration, suggesting that there are other unknown chemotactic molecules in the supernatants. During the *H. pylori* infection of the gastric mucosa, several bacterial ligands are produced, including lipopolysaccharide, peptidoglycane, secretory virulent proteins such as VacA (Vacuolating Cytotoxin A), etc., which results in the secretion of multiple chemokines and DAMPs by the gastric epithelial cells. Indeed, the CagA overexpression induced the migration of EBV-infected B lymphocytes but at lower numbers than the bacterial infection. Likewise, the bioinformatic analysis of the EBVaGC unveils an enrichment of multiple chemokines and receptors. Most probably, there are other ligand–receptor pairs also mediating the chemoattraction of the EBV-infected B lymphocytes to the site of the *H. pylori*-induced gastric inflammation. Further research is needed to identify all the players involved in the arrival of EBV to the gastric mucosa.

EBV is a lymphotropic virus that has also been associated with nasopharyngeal carcinoma (NPC) and GC, in addition to lymphomas. The main portal of entry and exit of the virus is the oral cavity, and EBV is frequently found infecting the B lymphocytes of the Waldeyer’s ring lymph nodes in the oro-naso-pharynx [19,25,69,70]. Viral reactivation in this anatomical site probably facilitates the viral transmission to new hosts, as well as the EBV-associated NPC, NKT cell lymphomas and other EBV-induced diseases of this anatomical site [24,71,72,73]. On the contrary, the stomach lacks the mucosa-associated secondary lymph nodes and it is not clear when and how the EBV arrives at this anatomical site and infects the epithelial cells.

*H. pylori* preferentially colonizes the gastric mucosa; it is proposed that the chronic inflammation induced by the infection is the main driver of GC. Several studies have documented the simultaneous presence of EBV and *H. pylori* in pre-neoplastic and neoplastic lesions [30,74,75]. A case-control serology study from our group found that the co-infection of EBV and pathogenic *H. pylori* CagA(+) strains is necessary to develop a severe inflammation and a higher risk to progress to advanced gastric lesions, such as chronic atrophic gastritis, intestinal metaplasia and GC of the intestinal type [29]. Likewise, we found that patients co-infected with *H. pylori* CagA(+) strains and EBV demonstrated a significantly higher association with severe inflammation in pediatric patients with non-atrophic gastritis than those infected with only one pathogen [30]. Altogether, these studies suggested that EBV, together with *H. pylori* CagA(+) strains, is an early participant in the chronic inflammation-mediated damage of the gastric mucosa that leads to the progression to advanced pre-neoplastic lesions and GC.

Other studies support EBV and *H. pylori* interactions, especially regarding the epigenetic regulation of the gastric epithelium, viral reactivation and synergy of their oncogenic potential. In the gastric epithelium infected by *H. pylori*, the oncogenic activity of CagA is downregulated by the SHP1 phosphatase. However, in co-infected cells, the EBV infection results in the epigenetic silencing of *PTPN6*, the gene that encodes SHP1, increasing the oncogenic potential of both pathogens [76]. A meta-analysis of GC with a methylator phenotype found an association with EBV and *H. pylori*, which has led to proposals for demethylating agents in the treatment of GC [77,78]. Pandey S. et al. observed in an in vitro co-infection model that *H. pylori* CagA(+) strains enhanced the oncogenic potential of EBV by upregulating the expression of DNA methyl transferases, which resulted in the silencing of tumor suppressor genes through hypermethylation [79]. Saxena A. et al. found a positive correlation between *H. pylori*-infected tissues and a high EBV load, suggesting reactivation mechanisms activated by the bacteria [80]. Consistent with this, Minoura-Etoh et al. found that *H. pylori* induces the production of monochloramine by gastric cells, which is an oxidant that promotes the reactivation of EBV [81].

IL-8 is a potent chemoattractant for neutrophils and other leukocytes [13], as well as an endothelial growth factor. Indeed, IL-8 induces angiogenesis and promotes the progression of different cancers, including GC [82,83,84]. Gastric *H. pylori* colonization has been associated with high IL-8 secretion by epithelial cells, particularly of CagA(+) strains [40,85]. A positive correlation has been observed between IL-8 levels and a high mucosal infiltration of neutrophils [14,16]. IL-8 also promotes the activation of neutrophils and other inflammatory cells [15], resulting in chronic gastric inflammation that persistently damages gastric tissue and leads to the advent and progression of gastric lesions [40,58,59,86,87]. Indeed, gastric inflammation with a high neutrophil infiltration is known as chronic active gastritis [88].

IL-8 secretion could explain the interaction of EBV with other pathogenic agents, such as *Porphyromonas gingivalis* or *Aggregatibacter actinomycetemcomitans*, two bacterial species that are involved in the development of periodontitis, a disease that arises as a result of the dysbiosis of the normal oral microbiota and that is also linked to EBV [89,90]. It has been demonstrated that infection by both *P. gingivalis* [91,92,93] and *A. actinomycetemcomitans* [93,94,95], in oral epithelial cell lines and primary cultures of the periodontal epithelium, induces the expression of IL-8, which could also explain the recruitment of EBV-infected B cells to the periodontal pocket. Also of interest, it is well recognized that EBV is present in 100% of the non-keratinizing NPC [96], and there is evidence that polymorphisms that enhance the IL-8 expression increase the risk of developing NPC [97,98]. More investigation is needed to address whether the overexpression of IL-8 at different anatomical sites may enhance the risk of EBV infection and pathogenesis.

IL-8 mainly signals through two receptors, CXCR1 and CXCR2, which are expressed on leukocytes and endothelial cells [99]. Upon activation, both receptors couple to pertussis toxin-sensitive G proteins to mediate the activation of phosphoinositide (PI)^2^ hydrolysis, intracellular Ca^2+^ mobilization, and MAPK and PI3K signaling pathways [100]. Although these receptors share extensive sequence homology, they only partially overlap in the use of ligands and functions. The CXCR2 expression has been linked to the recruitment of a variety of immune cells to the tumor microenvironment and to an unfavorable outcome in multiple types of cancer [101]. In GC, the CXCR2 expression was associated with poor tumor differentiation, increased tumor depth, lymph node metastasis and a short overall survival [65,102,103]. However, we did not observe a decreased survival of GC patients with high *CXCR2* expression in the TCGA database.

In summary, we propose a mechanism that may explain, at least partially, how EBV reaches the gastric epithelium. Our data support that *H. pylori*, which is the normal resident of the stomach, triggers an inflammatory microenvironment in the gastric mucosa, which is rich in IL-8. IL-8 secretion by the gastric epithelial cells is specially favored by infection with *H. pylori* CagA(+) strains. IL-8 attracts B lymphocytes to the site of infection, and migratory B lymphocytes are probably enriched with the CXCR2-expressing EBV-infected B cells and antigen-stimulated B cells, since we also observed an enhanced migratory response of the PMA-stimulated PBL. The IL-8 signals through CXCR2, activating the ERK1/2 and p38MAPK kinases, which are involved in processes such as cell migration, invasion, proliferation and survival [17,104]. Altogether, this mechanism of IL-8 chemoattraction may favor the infection of the gastric epithelial cells by EBV. In future studies, it is important to uncover those molecules responsible for the viral reactivation and production of infective particles and that facilitate the persistence of EBV in epithelial cells.

## Figures and Tables

**Figure 1 viruses-15-00651-f001:**
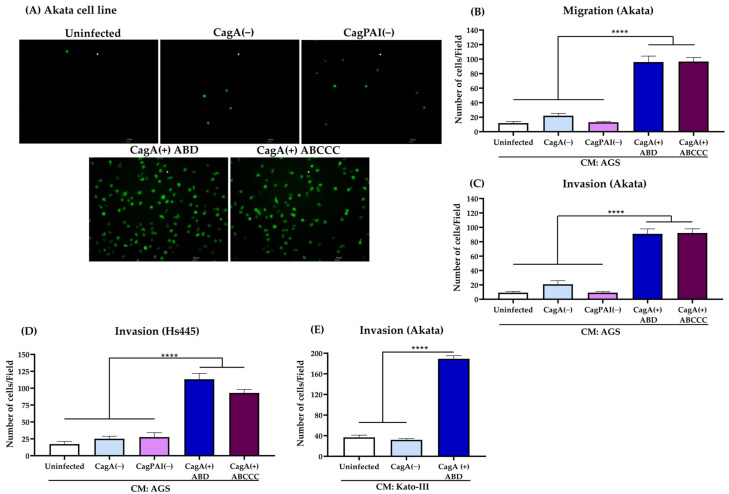
EBV-infected B lymphocytes are chemoattracted by components of conditioned media (CM) obtained from gastric epithelial cells infected with *H. pylori* strains. (**A**) Representative fluorescence images of Akata cells (EBV-GFP positive cells) chemoattracted to CM of AGS cells uninfected or infected with *H. pylori* CagA(−) or CagA(+) strains. The scale bars indicate 100 μm, magnification 10×. (**B**,**C**) Plots of the mean numbers of chemoattracted Akata cells in migration (**B**) and invasion assays (**C**). (**D**,**E**) Plots of invading Hs445 cells chemoattracted to CM from AGS cells infected with *H. pylori* strains, or invading Akata cells chemoattracted to CM from Kato-III infected with *H. pylori* strains. Graphs represent the mean ± SEM of the number of migrating or invading cells counted from five random microscope fields of three independent experiments. Non-parametric Kruskal–Wallis and Dunn’s post-hoc tests were used for the statistical analysis; **** *p* < 0.0001.

**Figure 2 viruses-15-00651-f002:**
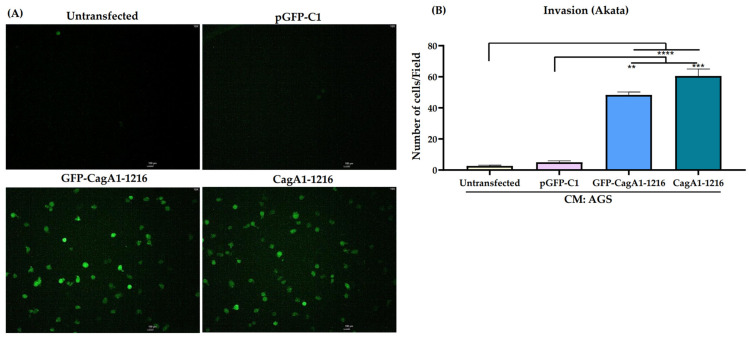
Overexpression of CagA in AGS cells chemoattracts EBV-infected B lymphocytes. (**A**) Representative fluorescence images of Akata cells (EBV-GFP positive cells) chemoattracted to conditioned media (CM) of AGS cells untransfected or transfected with empty vector (pGFP-C1), and full-length CagAs (GFP-CagA1-1216 or CagA1-1216). The scale bars indicate 100 μm with magnification of 10×. (**B**) Graph represents the mean ± SEM of the number of invading cells counted from 5 random microscope fields of three independent experiments. A non-parametric Kruskal–Wallis and Dunn’s post hoc tests were used for the statistical analyses. ** *p* < 0.01, *** *p* < 0.001 and **** *p* <0.0001.

**Figure 3 viruses-15-00651-f003:**
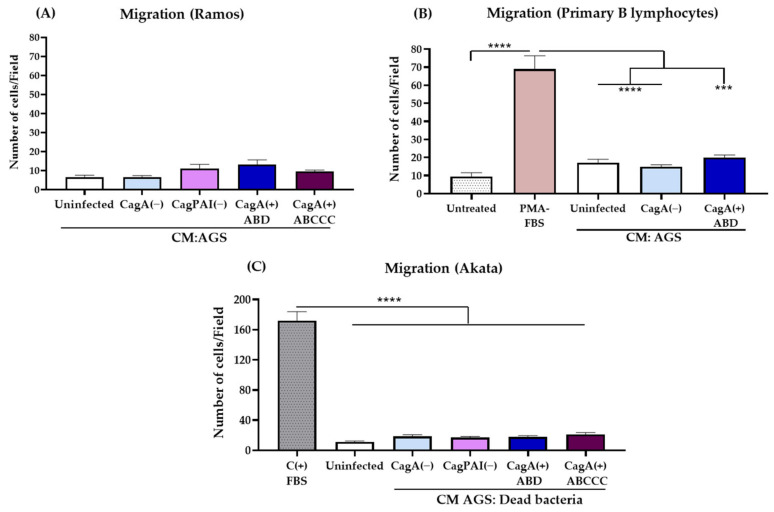
An active *H. pylori* and EBV infection is needed to chemoattract EBV-infected B lymphocytes. Migration assays of Ramos cells (**A**) and primary B lymphocytes (PBLs) (**B**) to conditioned media (CM) from AGS cells infected with *H. pylori* CagA(−) or CagA(+) strains. Appendix A show representative images of each assay (the scale bars indicate 100 μm with magnification of 10×). (**A**,**B**) Plots of the percentage of migrating cells. PBLs were purified from three healthy donors; data of the purity and viability of the PBLs are shown in the Appendix A. (**C**) Plots of the mean numbers of migrating Akata cells to CM from AGS cells infected with dead bacteria. FBS was used as positive control (C+) of chemoattraction of Akata cells and of PMA-stimulated PBLs. Graphs represent the mean ± SEM of the number of migrating cells counted from 5 random microscope fields of three independent experiments. A non-parametric Kruskal–Wallis and Dunn’s post hoc tests were used for the statistical analyses. *** *p* < 0.001 and **** *p* < 0.0001.

**Figure 4 viruses-15-00651-f004:**
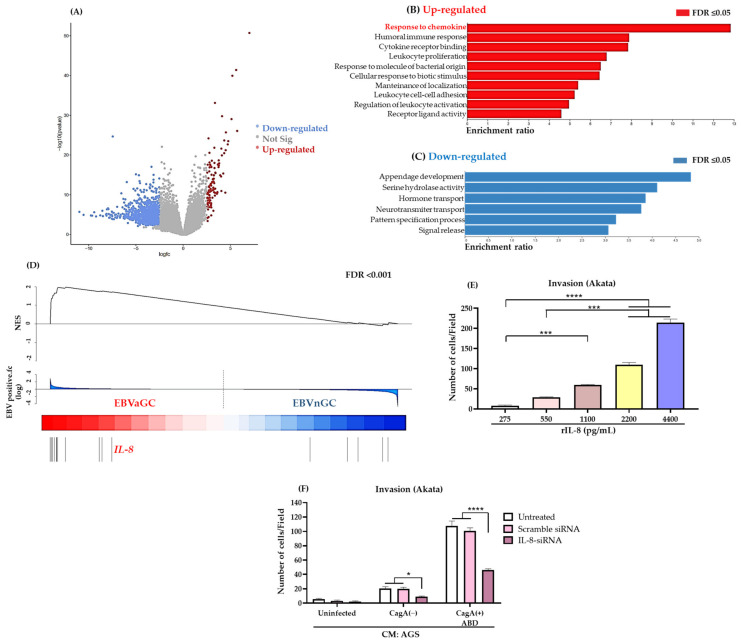
IL-8 is an important chemoattractor of EBV-infected B lymphocytes. (**A**) Volcano plot shows differentially expressed genes (DEG) between samples EBVaGC (*n* = 22) versus EBVnGC (*n* = 218). Red dots indicate up-regulated genes and blue dots indicate down-regulated genes, grey dots are genes that are not significantly different (Not sig). (**B**,**C**) Gene Ontology (GO) analysis of DEG in the comparison of EBVaGC vs. EBVnGC. Genes with a fold change up (**B**) or down (**C**) of at least 2.5 and False Discovery Rate (FDR) with a *p*-value **≤** 0.05 were included in this analysis. (**D**) Normalized Enrichment Score (NES) of a coined gene signature of chemokine and chemokine receptors (see Appendix A) to compare EBVaGC versus EBVnGC samples, *IL-8* (in red) appears as one of the chemokines with a top score. (**E**) Invasion assays of Akata cells chemoattracted to increasing concentrations of rIL-8. A non-parametric Kruskal–Wallis and Dunn’s post hoc tests were used for the statistical analyses. (**F**) Invasion assays of Akata cells chemoattracted to conditioned media from AGS cells knocked down in *IL-8* expression and then infected with *H. pylori* strains. As controls, cells were left untreated or transfected with a scramble siRNA. (**E**,**F**) Graphs represent the mean ± SEM of the number of invading cells counted from 5 random microscope fields of three independent experiments. A 2-way ANOVA for multiple comparison and Dunnett’s post hoc test were used to identify statistical differences. * *p* < 0.05, *** *p* < 0.001 and **** *p* < 0.0001.

**Figure 5 viruses-15-00651-f005:**
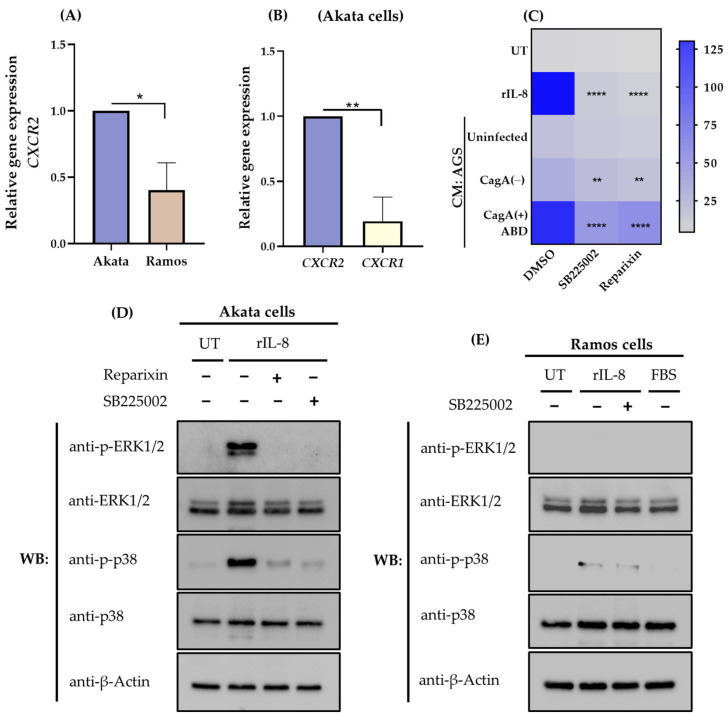
IL-8 promotes activation of ERK1/2 and p38MAPK mediated by CXCR2. (**A**) *CXCR2* relative expression in Akata and Ramos cells in basal conditions. (**B**) *CXCR1* and *CXCR2* relative gene expression in Akata cells. Graphs represent the mean ± SEM of the relative gene or fold-change expression of three independent experiments. Expression values were normalized to *GAPDH*, and unpaired t-test was used to assess statistical significance between groups. (**C**) Heatmap of invading Akata cells chemoattracted by either the conditioned media from AGS cells infected with *H. pylori* CagA(−) and CagA(+) ABD strains or rIL-8 (2200 pg/mL), with or without pretreatment with CXCR2 inhibitors (SB225002 and reparixin). Bar scale indicates the number of invading cells. Statistical differences were assessed comparing inhibitors versus vehicle (DMSO). A two-way ANOVA was used for multiple comparisons and a Tukey’s post hoc test. (**D**,**E**) Western blot analysis of phosphorylated p-ERK1/2, ERK1/2, p-p38MAPK, p38MAPK and β-Actin in Akata and Ramos cells treated or untreated (UT, control) with rIL-8 (2200 pg/mL), with or without pretreatment with the inhibitors. The images show one representative experiment of three independent assays (uncropped membranes are presented in Appendix A). * *p* < 0.05, ** *p* < 0.01 and **** *p* < 0.0001.

## Data Availability

Not applicable.

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
