# Peer review of "IL-8 Secreted by Gastric Epithelial Cells Infected with Helicobacter pylori CagA Positive Strains Is a Chemoattractant for Epstein–Barr Virus Infected B Lymphocytes"

_viruses, 2023, doi:10.3390/v15030651_

Round 1

Reviewer 1 Report

In the manuscript entitled "IL-8 Secreted by Gastric Epithelial Cells Infected with Helicobacter pylori CagA Positive Strains is a Chemoattractant for Epstein Barr-Virus Infected B Lymphocytes" the authors dissected a mechanism of how the EBV positive B lymphocytes were attracted to the H. pylori infected gastric epithelial cells. The authors pointed out IL8 functioning as a chemoattractant that binds to CXCR2 present on the B cells and attracts the B cells towards infected site.

The manuscript is well designed and has been drafted nicely. There are few concerns that needs to be addressed

1. The authors showed that the heat killed H. pylori cannot induced chemoattraction of AKATA cells. It will be interesting to see if exogeneous overexpression of the Cag in the AGS cells when infected with heat killed H. pylori can induce chemoattraction towards the AKATA cells.

2. the authors mentioned that the RAMOS cells expresses less CXCR2 compared to AKATA cells. It will be fascinating to see the consequence of overexpression of CXCR2 in RAMOS cells and whether during that condition the RAMOS cells behaves similarly to that of AKATA cells.

3. There are other infections that can also induce IL8. I wonder whether all these have the same consequence towards attracting EBV positive cells.

4. The quality of some of the figures are very poor. Specially the migration assay I can hardly see anything. Moreover the WB data are too overexposed and there is no visible background. Please reduce the brightness and contrast of these blots. 

Author Response

We truly appreciate the time and care of the reviewers who helped us improve the quality of the study. We now proceed to give a point-by-point answer to the reviewers´ comments. Changes to the original manuscript are highlighted either with the World´s function tracker of changes, and major changes to the specific reviewer´s observations are highlighted in yellow.

In the manuscript entitled "IL-8 Secreted by Gastric Epithelial Cells Infected with Helicobacter pylori CagA Positive Strains is a Chemoattractant for Epstein Barr-Virus Infected B Lymphocytes" the authors dissected a mechanism of how the EBV positive B lymphocytes were attracted to the H. pylori infected gastric epithelial cells. The authors pointed out IL8 functioning as a chemoattractant that binds to CXCR2 present on the B cells and attracts the B cells towards infected site.

The manuscript is well designed and has been drafted nicely. There are few concerns that needs to be addressed

  1. The authors showed that the heat killed H. pylori cannot induced chemoattraction of AKATA cells. It will be interesting to see if exogeneous overexpression of the Cag in the AGS cells when infected with heat killed H. pylori can induce chemoattraction towards the AKATA cells.

RESPONSE. Attending the reviewer concern, we performed an experiment in which full-length CagA was overexpressed in AGS cells, observing that CagA alone is sufficient to promote the release of IL-8, chemoattracting the Akata EBV-infected B cell line. This new data is shown in the new Figure 2 and Supplementary Figure 2.

  1. the authors mentioned that the RAMOS cells expresses less CXCR2 compared to AKATA cells. It will be fascinating to see the consequence of overexpression of CXCR2 in RAMOS cells and whether during that condition the RAMOS cells behaves similarly to that of AKATA cells.

RESPONSE. Although we agree that this experiment is very interesting, unfortunately, we do not have an easy and quick access to a plasmid encoding the full-length CXCR2. We also do not have the means to overexpress it in B lymphocytes, for example a lentivirus system, since lymphocytes are transfected very inefficiently. To perform this experiment would take us several months, and we do not believe that it will modify the conclusions we have already reached with the study.

  1. There are other infections that can also induce IL8. I wonder whether all these have the same consequence towards attracting EBV positive cells.

RESPONSE. Attending this comment, we added a paragraph in the discussion section (line 625 to line 636, of the document without track changes), where we mention that both P. gingivalis [PMID: 24887636, 11077999, 19778328] and A. actinomycetemcomitans [19778328, 9802708, 21545698] infection of oral epithelial cell lines and primary cultures of periodontal epithelium induce expression of IL-8, which could also recruit EBV-infected B lymphocytes to the periodontal pocket and explain the association between EBV and periodontal disease. We also mention that polymorphisms on IL-8 that enhance IL-8 expression increase the risk of developing NPC [PMID: 17869651, PMID: 17720627].

  1. The quality of some of the figures are very poor. Specially the migration assay I can hardly see anything. Moreover the WB data are too overexposed and there is no visible background. Please reduce the brightness and contrast of these blots.

RESPONSE. Attending the reviewer comment, the images of the chemoattraction assays were moved to the supplementary data (Supplementary Figure S1A and S3A,B ). There, we could enhance the size of the images to improve their sharpness. We have also reduced the brightness and contrast of the blots and converted the images to grayscale to improve the quality of the image (Figure 5D,E) and Supplementary Figure S7A,B).

Reviewer 2 Report

The current study by Domínguez-Martínez et al. highlights IL-8 Secreted by Gastric Epithelial Cells Infected with Helicobacter pylori CagA Positive Strains as an important Chemoattractant for Epstein Barr-Virus. The study is indeed interesting, as it highlights how infection with one pathogen can invite the other ultimately leading to worst pathology. However, it’s a preliminary study, having following issues:

There are no line numbers in the script that makes it hard for the reviewer to highlight anything.

In abstract: Check the spelling “parcially”.

In the first paragraph of introduction: Add the data mentioning global number of gastric cancer cases, number of cases suffering from EBV infection and percent cases of EBV resulting in gastric cancer.

Introduction: 2nd paragraph, 1st line: Check the English of the sentence.

Page 2: Line 2 to 4; Meaning is not clear. Please reframe the sentence.

Page 8: Line 4; Meaning is not clear. Please reframe it.

It is good that the authors have tried to conclude the information available in the public databases, but have not included any of the animal experiments. Thus, animal data confirming the findings of in-vitro experiments is not available. Inclusion of animal data will make it more realistic.

Author Response

We truly appreciate the time and care of the reviewers who helped us improve the quality of the study. We now proceed to give a point-by-point answer to the reviewers´ comments. Changes to the original manuscript are highlighted either with the World´s function tracker of changes, and major changes to the specific reviewer´s observations are highlighted in yellow.

Comments and Suggestions for Authors

The current study by Domínguez-Martínez et al. highlights IL-8 Secreted by Gastric Epithelial Cells Infected with Helicobacter pylori CagA Positive Strains as an important Chemoattractant for Epstein Barr-Virus. The study is indeed interesting, as it highlights how infection with one pathogen can invite the other ultimately leading to worst pathology. However, it’s a preliminary study, having following issues:

  1. There are no line numbers in the script that makes it hard for the reviewer to highlight anything.

RESPONSE. We have added the line numbers in the resubmitted version of the manuscript.

  1. In abstract: Check the spelling “parcially”.

RESPONSE. We have corrected this mistake.

  1. In the first paragraph of introduction: Add the data mentioning global number of gastric cancer cases, number of cases suffering from EBV infection and percent cases of EBV resulting in gastric cancer.

RESPONSE. We have included the following data in the revised version of the manuscript (1st paragraph of Introduction): 1.1 million cases and 750,000 deaths from gastric cancer per year worldwide. Also, 10% of gastric cancers have been associated with EBV infection.

  1. Introduction: 2nd paragraph, 1st line: Check the English of the sentence. Page 2: Line 2 to 4; Meaning is not clear. Please reframe the sentence. Page 8: Line 4; Meaning is not clear. Please reframe it.

RESPONSE. We have carefully read the text correcting grammar and style.

  1. It is good that the authors have tried to conclude the information available in the public databases, but have not included any of the animal experiments. Thus, animal data confirming the findings of in-vitro experiments is not available. Inclusion of animal data will make it more realistic.

RESPONSE. EBV is an exclusive parasite of humans, and to date there are not convenient models of infection of small animals. There are transcriptomic data generated in humanized mice (PMID: 32132242), in which PBMCs from human cord blood were infected with EBV and injected into NSG mice. Even though this could be considered as an animal model, the cells infected with EBV are human hematopoietic cells, while the epithelia remains of murine origin. On the other hand, there are H. pylori infection of murine and monkey cells. It has been found that H. pylori infection in mice (PMID: 35603777, 36364906) and primary cell cultures derived from Rhesus monkeys (PMID: 28813514) induces expression of IL-8, as it does in human cells. Establishing an animal model of EBV and H. pylori co-infection is not an easy task due to the restricted tropism of EBV.

Reviewer 3 Report

Both Helicobacter Pylori (H. Pilori) and Epstein-Barr virus (EBV) - a lymphotropic herpesvirus associated with several malignancies worldwide - have been found to be risk factors for the development of gastric cancer (GC). The authors previously suggested a correlation between coinfection with these two pathogens and severe gastric inflammation, however, the mechanims involved were not known.

In the present study, the authors demonstrate that infection of gastric epithelial cells by H. Pilori CagA positive virulent strains promotes EBV-infected B lymphocytes Chemoattraction. They then identified IL8 - one of the top cytokine whose expression increases during H. Pilori infection - as an important chemoattractant for EBV-infected B lymphocytes. Moreover, they found that expression of the IL8 receptor, CXCR2, is induced by EBV infection, enhancing the capacity of EBV-infected lymphocytes to be attracted by IL8 producing cells. Finally, they propose a mechanisms in which IL8 chemoattraction may favor infection of the gastric epithelial cells by EBV.

This is a solid and elegant study that brings novel informations on the possible mechanisms by which H. Pilori and EBV cooperate in the development of Gastric Cancer. Moreover, the manuscript is very well written and organised.  

Specific comments:

- In the summary, line 11, an s is missing in “expression” and line 13 partialy is written with a c instead of t.

- Could the authors enhance the photo panels of Figure 2. The cells are almost impossible to distinguish.

Author Response

We truly appreciate the time and care of the reviewers who helped us improve the quality of the study. We now proceed to give a point-by-point answer to the reviewers´ comments. Changes to the original manuscript are highlighted either with the World´s function tracker of changes, and major changes to the specific reviewer´s observations are highlighted in yellow.

Comments and Suggestions for Authors

Both Helicobacter Pylori (H. Pilori) and Epstein-Barr virus (EBV) - a lymphotropic herpesvirus associated with several malignancies worldwide - have been found to be risk factors for the development of gastric cancer (GC). The authors previously suggested a correlation between coinfection with these two pathogens and severe gastric inflammation, however, the mechanims involved were not known.

In the present study, the authors demonstrate that infection of gastric epithelial cells by H. Pilori CagA positive virulent strains promotes EBV-infected B lymphocytes Chemoattraction. They then identified IL8 - one of the top cytokine whose expression increases during H. Pilori infection - as an important chemoattractant for EBV-infected B lymphocytes. Moreover, they found that expression of the IL8 receptor, CXCR2, is induced by EBV infection, enhancing the capacity of EBV-infected lymphocytes to be attracted by IL8 producing cells. Finally, they propose a mechanisms in which IL8 chemoattraction may favor infection of the gastric epithelial cells by EBV.

This is a solid and elegant study that brings novel informations on the possible mechanisms by which H. Pilori and EBV cooperate in the development of Gastric Cancer. Moreover, the manuscript is very well written and organised.  

Specific comments:

  1. In the summary, line 11, an s is missing in “expression” and line 13 partialy is written with a c instead of t.

RESPONSE. We have corrected those mistakes.

  1. Could the authors enhance the photo panels of Figure 2. The cells are almost impossible to distinguish.

RESPONSE. Attending the reviewer observation, the images in Figure 2 corresponding to the Ramos cell line and PBL migration assays were enlarged and placed as Supplementary Figure 3A,B.

Round 2

Reviewer 1 Report

The authors responded to all the queries raised by the reviewers and to my understanding this manuscript is fit to publish in Cancers with minor spelling checks and syntax errors if any. 

Reviewer 2 Report

Dear Editor

The authors have incorporated all the suggestions and the script has improved. I recommend the acceptance.